# Progressive Damage Modelling of Fiberglass/Epoxy Composites with Manufacturing Induced Waves Common to Wind Turbine Blades

Jared W. Nelson[1], Trey W. Riddle[2], and Douglas S. Cairns[3]

[1]SUNY New Paltz, Division of Engineering Programs, New Paltz, NY USA
[2]Sunstrand, LLC, Louisville, KY USA
[3]Montana State University, Dept. of Mechanical and Industrial Engineering, Bozeman, MT USA

**Abstract.** Composite wind turbine blades are typically reliable; however, premature failures are often in regions of manufacturing defects. While the use of damage modelling has increased with improved computational capabilities, they are often performed for worst-case scenarios where damage or defects are replaced with notches or holes. To better understand and predict these effects, an effects of defects study has been undertaken. As a portion of this study, various progressive damage modelling approaches were investigated to determine if proven modelling capabilities could be adapted to predict damage progression of composite laminates with typical manufacturing flaws commonly found in wind turbine blades. Models were constructed to match the coupons from, and compare the results to, the characterization and material testing study presented as a companion. Modelling methods were chosen from established methodologies and included continuum damage models (linear elastic with Hashin failure criteria, user-defined failure criteria, non-linear shear criteria), a discrete damage model (cohesive elements), and a combined damage model (non-linear shear with cohesive elements). A systematic, combined qualitative/quantitative approach was used to compare consistency, accuracy and predictive capability for each model to responses found experimentally. Results indicated that the Hashin and combined models were best able to predict material response to be within 10% of the strain at peak stress and within 10% of the peak stress. In both cases, the correlation was not as accurate as the wave shapes were changed in the model, correlation was still within 20% in many cases. The other modelling approaches did not correlate well within the comparative framework. Overall, the results indicate that this combined approach may provide insight into blade performance with known defects when used in conjunction with a probabilistic flaw framework.

## 1 Introduction

The US Department of Energy sponsored, Sandia National Laboratory led, Blade Reliability Collaborative (BRC) has been tasked with developing a comprehensive understanding of wind turbine blade reliability (Paquette, 2012). A major component of this task is to characterize, understand, and predict the effects of manufacturing flaws commonly found in blades. Building upon coupon testing, outlined in the companion paper (Nelson et al., 2017), which determined material properties and characterized damage progression, three composite material defect types were investigated: porosity, in-plane (IP) waviness, and out-of-plane (OP) waviness. These defects were identified by an industry Delphi group as being common and deleterious

to reliability (Paquette, 2012). Significant research into effects of common composite laminate defects has been performed for both porosity (Wisnom et al., 1996; Baley et al., 2004; Costa et al., 2005; Huang and Talreja, 2005; Pradeep et al., 2007; Zhu et al., 2009; Guo et al., 2009) and fiber waviness (Adams and Bell, 1995; Adams and Hyer, 1996; Cairns et al., 1999; Niu and Talreja, 1999; Avery et al., 2004; Wang et al., 2012; Lemanski et al., 2013; Mandell and Samborsky, 2013).

The goal of this portion of the overall project was to establish analytical approaches to model progressive damage in flawed composite laminates consistently and accurately predict laminate response. Multiple cases for each flaw type were tested allowing for progressive damage quantification, material property definition, and development of many correlation points in this work. As outlined in the following sections, there have been two primary modelling approaches used to assess damage progression in composite laminates: Continuum Damage Modelling (CDM), and Discrete Damage Modelling (DDM). While

these methods are well established, there has been little work directly assessing predictive capabilities when applied to wind turbine blade laminates with defects.

## 1.1 Continuum Damage Modelling Background

Continuum Damage Modelling (CDM) is a "pseudo-representation" that does not explicitly model the exact damage but instead, updates the constitutive properties as damage occurs (Kachanov, 1986). This allows for the relating of equations to

heterogeneous micro-processes that occur during strain of materials locally, and during strain of structures globally, insofar as they are to be described by global continuum variables given their non-homogeneity (Talreja, 1985; Chaboche, 1995). Thus, for typical CDM as the model iterates at each strain level, the constitutive matrix is updated to reflect equilibrium damage. Then as damage occurs, the elastic properties are irreversibly affected in ways that are similar to those in a general framework of an irreversible thermodynamic process (Kachanov, 1986). This may take place by reducing the elastic properties ($E_1$, $E_2$,

$v_{12}$, $v_{32}$, and $G_{12}$) in the stiffness matrix ($C$) of the stress-strain relationship. Damage is not directly measurable from this approach, but may be estimated for the continuum by altering observable properties: strength, stiffness, toughness, stability, and residual life.

There are two crucial considerations when modelling damage: the failure theory and ways to account for the damage. Typical failure criteria such as the maximum stress, the maximum strain, Hashin (1980), Tsai–Hill (1968), and Tsai–Wu (1971) are

widely used because they are simple and easy to utilize (Christensen, 1997). In reviews by Daniel (2007) and Icardi (2007), wide variations in prediction by various theories were attributed to different methods of modelling the progressive failure process, the non-linear behavior of matrix-dominated laminates, the inclusion or exclusion of curing residual stresses in the analysis, and the utilized definition of ultimate failure. Camanho and Matthews (1999) achieved reasonable experimental/analytical correlation using Hashin's failure theory to predict damage progression and strength in bearing, net-

tension, and shear-out modes.

To account for damage, progressive damage models of composite structures range from the simple material property degradation methods (MPDM) to more complex MPDM that combine CDM and fracture mechanics (Tay et al., 2005). Implementing a ply discount method whereby the entire set of stiffness properties of a ply is removed from consideration if

the ply is deemed to have failed has been well established (Maimi et al., 2007). Typical examples of MPDM utilize a 2D progressive damage model for laminates containing central holes subjected to in-plane tensile or compressive loading which are directly compared to experimental findings (Blackketter et al, 1993; Gorbatikh et al., 2007).

MPDM schemes are often implemented through user-defined subroutines (Chen et al., 1999; Xiao and Ishikawa, 2002; Goswami, 2005; McCarthy et al. 2005; Basu et al., 2007). Credited with being the first in this direction, Chang and Chang (1987) developed a composite laminate in tension with a circular hole where material properties were degraded to represent damage. Failure criteria were defined based on the failure mechanisms resulting from damage: matrix cracking, fiber-matrix shearing, and fiber breakage. A property reduction model was implemented and the results agreed for seven (7) independent laminates. Later, Chang and Lessard (1991) performed similar work on damage tolerance of laminated composites in compression with a circular hole with similar results. These methods have been utilized for other conditions and have been used to develop a 3D analysis methodology based on the incorporating Hashin failure criteria into a similar logic (Evcil, 2008). By advancing to 3D, the error dropped down to 2.6% from as high as 30%. Others have continually built upon these accumulation CDM approaches giving them breadth across a wide variety of composite material, loading, and structural applications (Camanho et al., 2007; Liu and Zheng, 2008; Sosa et al., 2012; Su et al., 2015).

**1.2 Discrete Damage Modelling Background**

In contrast, a DDM physically models the actual damage as it would physically occur through the load profile, typically as local failure of the constituents to be more consistent with the physical damage. With DDM approaches, constitutive properties do not physically change in a continuum sense, rather, the degradation is a consequence of a local failure as it would occur within a structure. In development of DDM approaches, knowledge *a priori* of the damage location is very helpful, though the result is they are generally computationally more expensive.

While several different DDM methods exist (Rice, 1988; Moës and Belytschko, 2002; Krueger, 2004; Tay et al., 2005), cohesive elements were chosen for this study due to the ability to control failure initiation. The Dugdale–Barenblatt cohesive zone approach may be related to Griffith's theory of fracture when the cohesive zone size is negligible compared with other characteristic dimensions (Dugdale, 1960; Barenblatt, 1962). The intent of the cohesive zone is to add an area of vanishing thickness ahead of the crack tip to describe more realistically the fracture process without the use of the stress singularity utilized in linear elastic fracture mechanics (Rice, 1988). Barenblatt (1962) theorized that a cohesive zone, that is much smaller that the crack length, exists near the crack tip and has a cohesive traction on the order of the theoretical strength of the solid. In addition, the parameters defining size of the zone and traction at onset are independent of crack size and extremal loads. Finally, no stress singularity exists because stresses are finite everywhere including at the crack tip. It is important to note that energy dissipation is an intrinsic mechanism of fracture with the cohesive approach in contrast to classic continuum fracture mechanics.

Zero thickness elements are useful with laminated composites because they may be placed between layers or fibers allowing Cui and Wisnom (1993) used this type of element to predict delamination progression in specimens under three-point bending

and in specimens with cut central plies. Duplicate nodes were used along the interface between distinct plies connected by two independent, zero thickness springs, horizontal and vertical. As expected, the cohesive elements used showed a sudden discontinuous change in stiffness when the failure criterion was reached. The method was further developed by creating an element that provided a smoother transition from linear elastic behavior to plastic behavior (Wisnom, 1996; Petrossian and Wisnom, 1998). Later, a quasi-3D model was proposed to predict, with reasonable results, both delamination and intra-ply damage prior to ultimate failure in a cross-ply laminate with a center crack loaded in tension (Wisnom and Chang, 200). Planar elements were used on the surface of each ply and were then connected with non-linear springs, as above, to model delamination between different plies. A similar technique was used to model longitudinal splitting along the fibers by means of spring interface elements across the line perpendicular to the notch where splitting is expected. A bi-linear traction-separation criterion is commonly employed such that the element has a linear stiffness response until the maximum traction point is reached and damage is initiated (Turon et al., 2007). Then, the second portion of the bi-linear response estimates the damage evolution up to failure where separation occurs and the element is deleted. While the cohesion properties may successfully be calculated (Sørensen and Jacobsen, 2003; Turon et al, 2007), use of cohesive elements has also been successful where the bi-linear response has been developed iteratively using experimental/analytical correlation (Tvergaard and Hutchinson, 1996; Allen and Searcy, 2001). While this method is computationally expensive due to extensive number of elements needed, this method has been widely shown to effectively model crack propagation.

## 2 Modelling Techniques

Several different modelling approaches were utilized to most accurately model the experimentation outlined in the testing companion paper (Nelson et al., 2017). It is important for the reader to note that all references to material testing and experimental results are from the work outlined in the companion paper. For each modelling approach, the geometry was set up to match the intended coupon size (100 mm x 50 mm) of the 4-layer uni-directional fiberglass used during material testing. Two-dimensional models were generated with both non-wave and wave geometries (Figure 1), with quadrilateral, plane strain elements (CPS4), in Abaqus where each element was generated to be consistent with the nominal fiber tow width (1.0 mm). An unflawed case was tested for each method using a fiber misalignment angle of 0° as verification of material properties and model setup. Porosity was also modelled with no fiber wave and material properties were degraded based on results from experimentation and matrix continuum degradation because of the porosity. The initial IP wave modelled had an amplitude (A) of 3.8 mm, a wavelength (λ) of 47.6 mm, and average off-axis fiber angle of 28.7°. Similarly, the initial OP wave modeled had an amplitude (A) of 2.9 mm, a wavelength (λ) of 22.8 mm, and average off-axis fiber angle of 29°. These variables were adjusted to match additional waves tested. Local coordinate systems were defined for the elements oriented to form the wave such that the fiber direction remained consistent through the wave. Since a symmetric wave was modelled, the number of elements was reduced using a symmetry boundary condition at the peak of the wave, as shown by the line of symmetry in Figure 1 to reduce processing time. The elements edges along the line of symmetry were fixed vertically (1-direction), but

were not constrained otherwise. A displacement condition was applied at the bottom to match the applied load during testing. Elastic material properties and damage progression determined in the coupon testing were utilized for all modelling methods, as shown in Table 1. The only variation of these empirically derived properties was with a reduction based on Kerner's approach used for the 2% porosity case (Kerner, 1956). However, in cases where model properties were found parametrically (but still consistent with test data), they may have been modified to optimize correlation as explained for each case below. After solving, symmetry was applied to allow for calculation of the full-field average strain and stresses for comparison to the experimental testing.

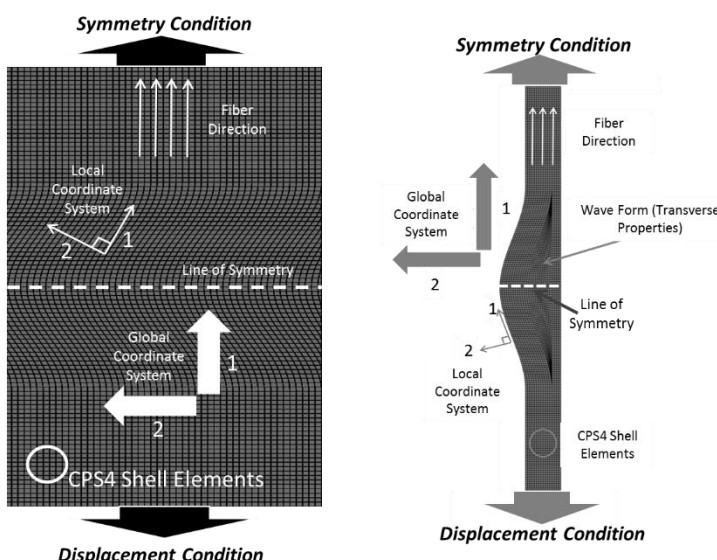

**Figure 1: Representation of model and references used for IP (left) and OP (right) wave model.**

**Table 1: Empirical material properties utilized in Progressive Damage Analysis.**

| | Longitudinal Modulus (GPa) | Transverse Modulus (GPa) | Poisson's Ratio | Shear Modulus (GPa) |
|---|---|---|---|---|
| | $E_1$ | $E_2$ | $v_{12}$ | $G_{12}$ |
| Tension | 40.6 | 16.3 | 0.27 | 16.8 |
| Compression | 38.4 | 14.4 | 0.28 | 14.4 |

Several assumptions were made to simplify this modelling effort. First, it was assumed that all fibers were parallel and uniform in the intended direction with reference to the width-wise edge, including through the wave. It was also assumed that all the fibers, for both the unflawed and wave geometries, were parallel and aligned through the thickness. These assumptions greatly simplified the modelling approach even though they were a possible source of the variation noted within the testing. In addition, perfect bonding between the layers was assumed.

## 2.1 Hashin-based Progressive Damage

The Abaqus built-in a Progressive Damage and Failure for Fiber-Reinforced Materials (Abaqus, 2012) that is intended to be used for elastic-brittle, anisotropic materials based on the Hashin failure criteria was utilized. In this case, the elastic response is defined as a linear elastic material with a plane stress orthotropic material stiffness matrix. However, damage initiation must also be defined for the four included mechanisms: fiber tension, fiber compression, matrix tension, and matrix compression. Damage is initiated when one or more of these mechanisms reaches a value of 1.0 or larger based on the material strengths shown in Table 2:

Fiber tension ($\hat{\sigma}_{11} \geq 0$):

$$F_f^t = \left(\frac{\hat{\sigma}_{11}}{X^T}\right)^2 + \alpha\left(\frac{\hat{t}_{12}}{S^L}\right)^2 \tag{1}$$

Fiber compression ($\hat{\sigma}_{11} < 0$):

$$F_f^c = \left(\frac{\hat{\sigma}_{11}}{X^C}\right)^2 \tag{2}$$

Matrix tension ($\hat{\sigma}_{22} \geq 0$):

$$F_m^t = \left(\frac{\hat{\sigma}_{22}}{Y^T}\right)^2 + \left(\frac{\hat{t}_{12}}{S^L}\right)^2 \tag{3}$$

Matrix compression ($\hat{\sigma}_{22} < 0$):

$$F_m^c = \left(\frac{\hat{\sigma}_{22}}{2S^T}\right)^2 + \left[\left(\frac{Y^C}{2S^T}\right)^2 - 1\right]\frac{\hat{\sigma}_{22}}{Y^C} + \left(\frac{\hat{t}_{12}}{S^L}\right)^2 \tag{4}$$

where $X^T$ is the longitudinal tensile strength, $X^C$ is the longitudinal compressive strength, $Y^T$ is the transverse tensile strength, $Y^C$ is the transverse compressive strength, $S^L$ is longitudinal shear strength, and $S^T$ is transverse shear strength. In addition, the shear stress contribution coefficient, $\alpha$, was set to be equal to 1 as done by Hashin (1980). Elemental properties were then degraded per the defined damage parameters in Table 2. In the case of the fracture energies necessary for damage evolution values were approximated utilizing an approximated area under the stress-strain curves. The damage evolution parameters were not found experimentally but instead longitudinal and transverse moduli of elasticity (Table 1) were utilized in tension and compression, respectively, with the respective tensile and compressive strengths (Table 2) assuming a brittle material response. Since these values were not found experimentally, it was determined that modification could be performed to improve correlation and discussion of such modifications is found below. Also, it was determined that the damage initiation parameters could be modified within 10% of the experimentally derived values as a consequence of the variations noted in the testing. Also, the damage evolution parameters should not be confused with the traditional strain energy release rate in fracture mechanics. The damage evolution values are the total strain energies dissipated for a given progressive damage process. Since this method is built-in to Abaqus, the reader is referred to the Abaqus Analysis User's Manual section on Damage and Failure for fiber-reinforced composites (Abaqus, 2012). Please note that this work is not intended to be a comprehensive study on the Hashin Failure Criterion implemented into ABAQUS. The Hashin Failure Criterion was used tor comparisons to the original work presented herein.

**Table 2: Damage initiation and evolution parameters utilized in Progressive Damage Analysis.**

| Property | Damage Initation (Strength) Parameters | | | | | | Damage Evolution (Energy Dissipation) Parameters | | | |
|---|---|---|---|---|---|---|---|---|---|---|
| | Longitudinal Tensile (MPa) | Longitudinal Compressive (MPa) | Transverse Tensile (MPa) | Transverse Compressive (MPa) | Longitudinal Shear (MPa) | Transverse Shear (MPa) | Fiber Tension (J/m²) | Fiber Compression (J/m²) | Matrix Tension (J/m²) | Matrix Compression (J/m²) |
| Symbol | $X^T$ | $X^C$ | $Y^T$ | $Y^C$ | $S^L$ | $S^T$ | $G^c_{ft}$ | $G^c_{fc}$ | $G^c_{mt}$ | $G^c_{mc}$ |
| Value | 990 | 582 | 60 | 35 | 112 | 124 | 1.29E+06 | 7.57E+05 | 7.80E+04 | 4.55E+04 |

## 2.2 User-defined Subroutine

Next, a user-defined subroutine was employed with a combined maximum stress/strain user specified failure criteria where the standard input file builds and meshes the model, while the user subroutine checks for damage at each step. If damage was detected, the material properties were adjusted as described in Table 3 or the loop was stopped if ultimate failure occurred. If damage was detected, but not ultimate failure, the material properties were degraded depending on the type of failure as outlined in Table 3 based on the three independent failure types: matrix cracking, fiber-matrix damage, and fiber failure. Based on the procedural logic from Chang and Chang (1987), an Abaqus code was written with a FORTRAN subroutine acting as the inner loop following the decision tree shown in Figure 2 (Chang and Lessard, 1991).

**Table 3: Progressive Damage Analysis degradation for User Defined Criteria**

| Material Failure Type | Elastic Property Adjustments for Each Failure Type | | | | Notes |
|---|---|---|---|---|---|
| No Failure | $E_x$ | $E_y$ | $v_{xy}$ | $G_{xy}$ | Full properties. |
| Matrix Cracking Damage | $E_x$ | 0 | 0 | $G_{xy}$ | Used in tensile and compressive cases. |
| Fiber-Matrix Damage | $E_x$ | $E_y$ | 0 | 0 | Fiber compresses & matrix cracks; used in compression only. |
| Fiber Failure | 0 | 0 | 0 | 0 | Fiber buckles or breaks; all properties drop to zero. |
| Combined Matrix Cracking & Fiber-Matrix Damage | $E_x$ | 0 | 0 | 0 | Fiber is still intact and able to carry some longitudinal load. |
| Combined Matrix Cracking Damage & Fiber Failure | 0 | 0 | 0 | 0 | All properties drop to zero. |
| Combined Fiber-Matrix Damage & Fiber Failure | 0 | 0 | 0 | 0 | |
| All Combined Failure Modes | 0 | 0 | 0 | 0 | |

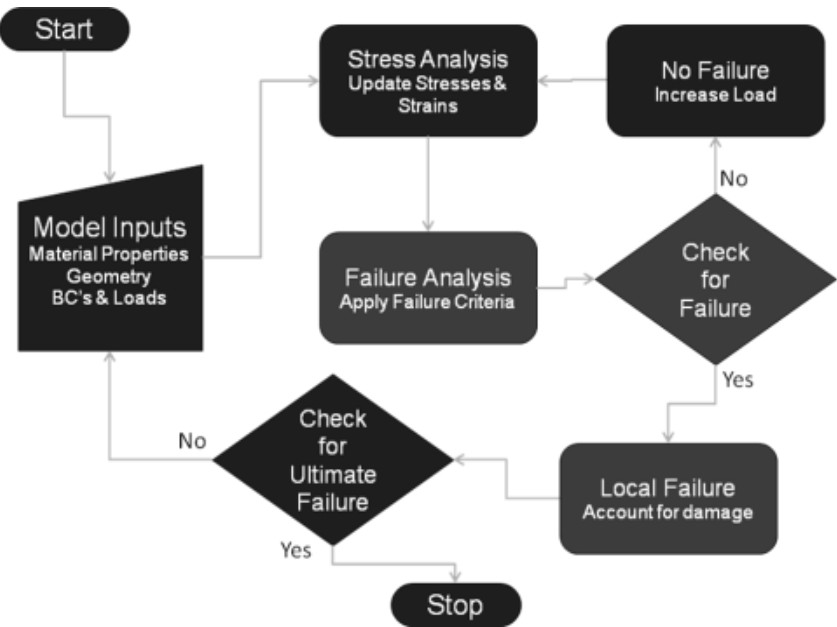

**Figure 2: Decision tree for progressive damage modelling utilized in this modelling.**

To determine the failure values, both maximum stress and strain criteria were implemented into the subroutine utilizing the material properties in Table 1, the damage initiation values in Table 2, and a strain at failure of 2.6%. If necessary, it was determined that the damage initiation parameters could be modified within 10% of the experimentally derived values given the variations noted in the testing to improve correlation. A modified maximum stress failure criterion was implemented with the inclusion of a maximum strain criteria to accurately model ultimate matrix failure. As such, matrix cracking damage was estimated by:

$$\left(\frac{\sigma_{22}}{Y_T}\right)^2 + \left(\frac{\tau_{12}}{S_T}\right)^2 = 1 \tag{5}$$

where $\sigma_{22}$ and $Y_T$ are transverse stress and transverse strength, respectively, and $\tau_{12}$ and $S_T$ are shear stress and strength, respectively. It must be noted that this same equation was utilized for both tensile and compressive cases, and the associated material properties were changed for each case. While the fiber-matrix compression damage case appeared to be necessary only in compression loading cases, with the given geometries these failure criteria were utilized in both tensile and compressive cases:

$$\frac{\sigma_{11,C}}{Y_C} + \frac{\tau_{12}}{S_T} = 1 \tag{6}$$

where $\sigma_{11,C}$ and $Y_C$ are fiber compressive stress and strength, respectively. Finally, two different equations were manually utilized depending on whether fiber failure is in tension or compression, respectively:

$$\frac{\varepsilon_{11,T}}{\bar{\varepsilon}_T} = 1 \qquad\qquad (7)$$

$$\frac{\sigma_{22,C}}{X_C} = 1 \qquad\qquad (8)$$

where $\varepsilon_{11,T}$ and $\bar{\varepsilon}_{11,T}$ were calculated for ultimate tensile strain and compressive stress, respectively. Utilization of the maximum strain criterion in tension was based on the consistency of strain at failure of these materials as determined in the testing. Integration of this criterion was a fundamental motivation in utilizing this user-defined technique.

A standard Abaqus code was written for an elastic material with 3 dependencies to match the independent failure types before calling out a *USER DEFINED FIELD* to call the subroutine into use. The subroutine itself was rewritten from the FORTRAN example found in the Abaqus Example Problem 1.1.14 and the reader is referred to this reference directly for the code specifics (Abaqus, 2012). First, the subroutine established the specific material parameters taken from the experimentation as noted in Tables 1 and 2 above. Next, the failure variables were initialized and the stresses were retrieved from the previous increment. Next, the crucial portion of the code was reached where the stresses were used to check for failure in each of the cases, or each dependent variable is determined. An *IF* loop was utilized for each of the Equations 5-8 noted, with 7 and 8 being manually swapped out for tension and compression, respectively. For example, considering the matrix damage portion, the loop first determined that if the matrix cracking damage variable was less than one (Table 3), the loop then recalculated with the updated stresses before updating the appropriate state variable. No calculation was necessary if the value was one because failure already occurred. Finally, the state variables were used to update the field variables which were then passed back to the standard code, and the loop was ended.

Thus, at each increment the subroutine ran through the failure criteria equations that analyze the stress and strain data of that increment. Resulting values of these equations range from zero (0) to one (1) with failure occurring when the value was equal to one (1). As the failure indices were calculated to be one (1), failure occurred in that element and the material properties were adjusted based on the failure type as noted in Table 3. For example, if a matrix failure occurred, the failure indices included in the user subroutine calculated that Failure Value #1 became equal to one (1). Thus, the elastic properties for that element only include $E_x$ and $G_{xy}$ as these are fiber dominated. The loop continued with the degraded properties until fiber failure or a combination of failures occurred resulting in no material properties for that element.

### 2.3 Non-linear Shear Model

Based on the shear between the fiber tows in the wavy area, it was deemed that a non-linear constitutive law needed to be developed for the bulk material by developing and using a user defined material subroutine (UMAT) in Abaqus (2012). As observed in the experimental testing and indicated by VanPaepegem et al. (2006), unrecoverable damage or plasticity occurs through the shear response. A method to degrade the shear material properties based on the shear response generalizing this damage and plasticity was implemented. Based on the change in shear modulus during this degradation, 8 points were

identified where changes in secant modulus were noted as identified in Figure 3. Otherwise, all material parameters were consistent with those listed in Table 1. The tabulated shear stress-strain relationships (Figure 3) were used to determine the shear stress and tangential modulus by the subroutine once the stress for the increment was calculated:

```
         SUBROUTINE UMAT_SHEAR_STIF(SHEAR,GAMMA,TAU,GG12,G12,STRESS3)
IMPLICIT REAL*8(A-H,O-Z)
              DIMENSION GAMMA(*),TAU(*),GG12(*)
              IF   (SHEAR.LT.GAMMA(1)) THEN
                 G12=GG12(1)
                 STRESS3=G12*SHEAR
ELSEIF(SHEAR.LT.GAMMA(2)) THEN
                 G12=GG12(2)
                 STRESS3=TAU(1)+G12*(SHEAR-GAMMA(1))
              ELSEIF(SHEAR.LT.GAMMA(3)) THEN
              ⋮
{SIMILAR ELSEIF STATEMENTS CONTINUE FOR THE NEXT 4 STRESS LEVELS}
              ⋮
              ELSE
                 G12=GG12(8)
                 STRESS3=TAU(8)+G12*(SHEAR-GAMMA(8))
ENDIF
              RETURN
              END
```

Once the shear stress and modulus were determined, the updates were returned into the material card of the model. It is important to note that since the tabulated shear points were identified as points where the slope of the curve changed dramatically, correlation might be improved by taking other points so long as they were from the same data set. In other words, it was determined that the tabulated points could be changed to potentially improve correlation.

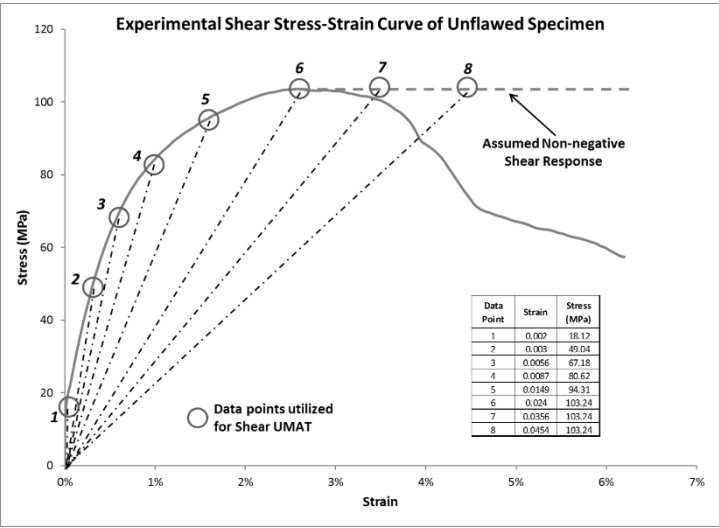

**Figure 3: Key points from empirical shear stress-strain relationship used in the non-linear shear UMAT.**

## 2.4 Cohesive Zone Model

To model damage progression discretely, cohesive elements are typically utilized based on a cohesive law relating traction to separation across the interface (Karayev et al., 2012; Lemanski et al., 2013). Zero thickness elements with specific bi-linear traction-separation criteria (Figure 5, right) were placed between the fiber tows of the material properties in Table 1 above. While following convention to utilize cohesive elements only in specific areas, computational availability has made it conceivable to place cohesive elements between all fiber tows throughout the model. Thus, damage and crack progression could occur between any fibers based on the stress state. It is important to note that the damage does not necessarily occur at the cohesive zone area. It only provides the opportunity for growth where damage can, and has been experimentally determined to grow before final failure. Damage growth only occurs when and where the critical load is met. This is an important distinction from assuming a damage path as in the case of conventional Linear Elastic Fracture Mechanics.

A bi-linear traction-separation criterion was implemented (Figure 4) where the initial stiffness, $K$, of the cohesive element is linear up to the damage initiation point at critical separation, $\Delta_c$. From this point to the failure separation, $\Delta_{fail}$, the slope estimates the damage evolution of each the cohesive element up to failure. The traction-separation criterion is met for a specific cohesive element and a separation occurs resulting in crack propagation and element deletion. A standard material specification was used and parametric studies were performed to determine the stiffness and maximum traction properties of the cohesive elements (Figure 5). Given these parametric studies, it was deemed that these values may be modified if necessary within the ranges determined from the study. Initial model analyses were performed to determine the cohesive element stiffness, $K_{eff}$. Analyses were performed at various stiffness values to determine the convergence value of 5E6 N/mm by performing several model runs to determine convergence point (Figure 5a). Similarly, the effects of $T_{1max}$ were determined by analyzing several different values and it was determined that failure behavior was not dependent on $T_{1max}$ (Figure 5b). However, when a similar test was analyzed for $T_{2max}$ it was quickly apparent that the failure was sensitive to Mode II shear damage (Figure 5c). The peak tractions ($T_{1max} = T_{2max} = 100$ MPa) were then used in an initial run as shown in Figure 5d to confirm these values and ensure that the cohesive elements were not influencing initial stiffness correlations. A B-K mixed mode criterion was utilized where $G_{Ic}$ and $G_{IIc}$ were 806 J/m$^2$ and 1524 J/m$^2$, respectively, as found experimentally (Benzeggagh and Kenane, 1996).

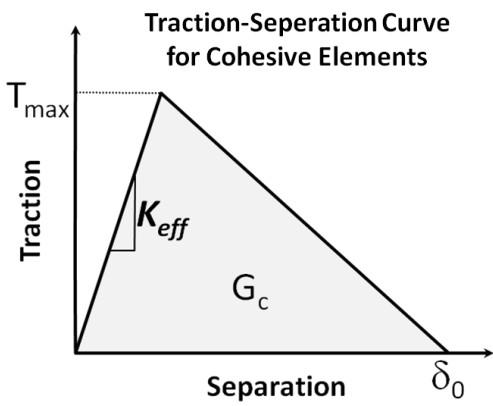

**Figure 4: Representation of bi-linear traction-separation response for a cohesive element.**

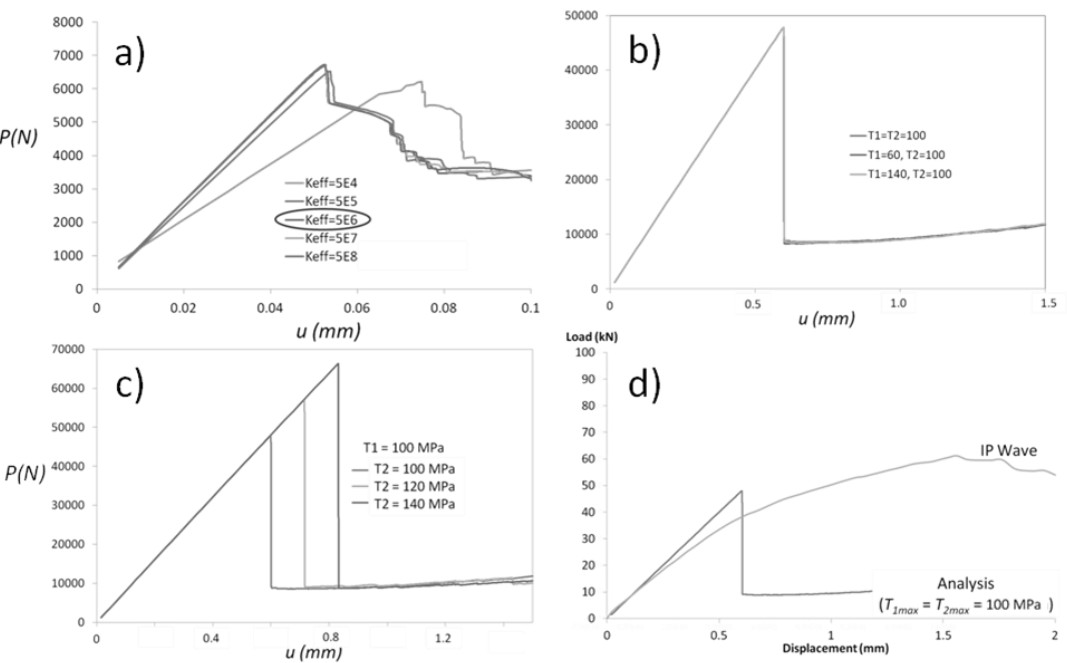

Figure 5: Results of parametric studies to find cohesive element: a) effective stiffness, $K_{eff}$; b) peak Mode I traction, $T_{1max}$; c) peak Mode II traction, $T_{2max}$; and, d) confirmation of peak tractions.

## 2.5 Combined Non-linear Shear and Cohesive Zone Model

The non-linear shear CDM and the DDM using cohesive elements were combined due to their poor overall performance individually. As discussed below, in both cases, the models seemed to capture portions damage progression, while each lacked the exact progression observed in the testing. In this case, the methods described in Sections 2.3 and 2.4 above, were combined

by adding the non-linear shear routine to the cohesive zone model with the same material properties and parameters utilized from the material testing and parametric studies performed.

## 3 Model Validation Methodology

A systematic approach, as shown in Figure 6, was employed to validated and compare different modelling methods. A qualitative/quantitative approach was utilized similar to that utilized by Lemanski et al. (2013), though strains at peak stress were also considered. The extensive test program found in the companion paper was used to validate this work both qualitatively and quantitatively (Nelson et al., 2017). As such, acceptable models correlated well both qualitatively, by matching failure location and shape, and quantitatively, by matching initial stiffness and peak stress at failure strain, to these experimental results. First, a qualitative assessment was performed and correlation was deemed acceptable if strain accumulation and damage progression visually matched the testing results. Using digital image correlation results from the material testing allowed for quick analysis of several key factors including an energy comparison. An energy comparison ensured that the energy was conserved between the strain energy available and energy dissipated. A visual comparison of the unrecoverable energy, or area under the curves, was deemed sufficient as models that do not conserve energy were evident and were not considered acceptable.

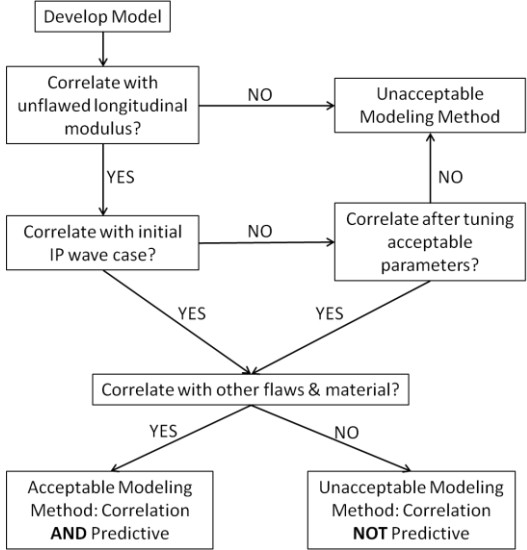

**Figure 6: Systematic flow of approach to determine acceptability of each model.**

If the qualitative criteria were met, a quantitative assessment was performed. First, the strain at peak stress was compared and deemed acceptable if it was within ±10% of testing results. If acceptable, peak stress was compared and deemed acceptable if it was within ±10% of testing results. The value of 10% was chosen for both parameters as it was the smallest range of all the experimental variability as shown in Figure 7. While these acceptance criteria were beyond the variability noted in the

testing, if these criteria were outside ±10%, but within ±20%, correlation was considered moderate and model modification was performed. It is important to note that this consideration was only made for correlation with other flaws after acceptable correlation had been achieved for the initial IP wave case. As such, models were considered predictive if correlation was achieved with these other cases utilizing the same input parameters as the initial IP wave case.

**Figure 7: Tension and compression response of IP Wave 1 utilized for baseline correlations with associated experimental variability.**

As shown in Figure 6, if correlation was not achieved by a model at any point during the systematic increase in flaw complexity, the model was deemed unacceptable and no additional flaw geometries were tested. The increase in flaw complexity in each case progressed from unflawed controls to porosity to the IP wave baseline case (Figure 1, left) to the initial OP wave case (Figure 1, right), and then to other IP and OP geometries. Acceptable models were able to accurately and consistently predict each of these cases, and with this consistent systematic approach, the different techniques were compared. The analytical models presented above were created, run, and correlated to responses outlined in the testing effort and modified, if necessary, to improve correlation if found to be outside the ±10% indicated above. As noted individually in Sections 2.1-2.5 and shown in Table 4, the specific input parameters for each model are shown as well as the parameters that were acceptable to tune within the ranges of the variability seen during experimentation. Acceptable tuning parameters within the variability noted from the experimental results was performed only to assist with convergence, and the effects on the model were directly tracked as discussed below. It is critical to note that no results included were from modifications made to any elastic properties shown in Table 1.

**Table 4: Input parameters and acceptable parameters for modification with range of acceptable modification.**

| MODEL TYPE | MODEL | INPUT PARAMETERS | PARAMETERS ACCEPTABLE FOR MODIFICATION |
|---|---|---|---|
| CDM | Linear Elastic | ELASTIC PROPERTIES | NONE |
| | Linear Elastic w/ Hashin Failure Criteria | ELASTIC PROPERTIES & DAMAGE INITIATION & EVOLUTION | DAMAGE INITIATION & EVOLUTION PARAMETERS (TABLE 2) |
| | Subroutine w/ User defined Damage Criteria | ELASTIC PROPERTIES, DAMAGE INITIATION, & FAILURE CRITERIA | DAMAGE INITIATION (TABLE 2) |
| | Non-Linear Shear | ELASTIC PROPERITES & STRESS-STRAIN FROM UNFLAWED SHEAR RESPONSE | ADJUSTMENT OF POINTS FROM SHEAR STRESS-STRAIN RESPONSE (FIGURE 3) |
| DDM | Cohesive Elements between Tows | ELASTIC PROPERTIES & COHESIVE TRACTION-SEPARATION | COHESIVE TRACTION-SEPARATION (PARAMETRICALLY DETERMINED IN FIGURE 5) |
| Combined | Non-Linear Shear w/ Cohesive Elements between Tows | ELASTIC PROPERTIES, STRESS-STRAIN FROM UNFLAWED SHEAR RESPONSE, & COHESIVE TRACTION-SEPARATION | ADJUSTMENT OF POINTS FROM SHEAR STRESS-STRAIN RESPONSE (FIGURE 3) & COHESIVE TRACTION-SEPARATION (PARAMETRICALLY DETERMINED IN FIGURE 5) |

## 4 Results & Discussion

The results from each model following the validation methodology are summarized in Table 5. These results are discussed through the progression of increasing complexity (unflawed, porosity, IP wave, OP wave, and additional waves, respectively) for each model. When compared to the experimental results, each model was scored based on the acceptance criteria with acceptable correlation (A), moderate correlation (M), and unacceptable correlation (U). There were several cases where experimental results were not yet available due to complexity of testing (R). Also, once a method was deemed unacceptable no additional models were run through the increasing complexity (NR). It should be noted that a modulus check (MC) on the unflawed specimen confirmed modulus correlation.

**Table 5: Summary of results of each model for acceptability.**

| MODEL TYPE | MODEL | UNFLAWED | | POROSITY | | IP WAVE | | OP WAVE | | ADDITIONAL WAVES | |
|---|---|---|---|---|---|---|---|---|---|---|---|
| | | Tension | Comp | Tension | Comp | Tension | Comp | Tension | Comp | Tension | Comp |
| CDM | Linear Elastic | MC | MC | NR | NR | NR | NR | NR | NR | NR | NR |
| | Linear Elastic w/ Hashin Failure Criteria | MC | MC | A | M | A | M | M | R | A,U | NR |
| | Subroutine w/ User-defined Damage Criteria | MC | MC | NR | NR | U | U | NR | NR | NR | NR |
| | Non-Linear Shear | MC | MC | NR | NR | U | U | NR | NR | NR | NR |
| DDM | Cohesive Elements between Tows | MC | MC | NR | NR | U | U | NR | NR | NR | NR |
| Combined | Non-Linear Shear w/ Cohesive Elements between Tows | MC | MC | NR | NR | A | A | A | R | A,U | R |

KEY:
A = ACCEPTABLE CORRELATION (visual correlation and within 10% of Strain at Peak Stress & within 10% of Peak Stress)
M = MODERATE CORRELATION (visual correlation but marginal quantitative acceptance criteria)
U = UNACCEPTABLE CORRELATION (unacceptable visual and/or quantitative correlation)
R = MODEL RUN BUT NOT CORRELATED (insufficient test data available)
NR = MODEL NOT RUN (due to unacceptable initial case or acceptable overall method)
MC = INITIAL MODULUS CHECK (stiffness of model within 5% of test)

## 4.1    Unflawed and Porosity Correlations

For each modelling technique, a qualitative analysis, and then a quantitative analysis, was performed. The impetus of this was
to ensure that the progressive damage models were consistent with the observed progressive damage in tests. The preliminary
step for each model case was ensure the unflawed material response matched experimental results.  Given the simplicity of the
check, only a qualitative comparison of the initial modulus was made. In all cases, correlation was found to be within 5% as
shown for a representative case in Figure 8, left. A similar result is noted for the 2% porosity case correlation for the linear
elastic with Hashin failure criteria (Figure 8, right).  Given the good correlation between this method and the ease-of-use with
the Kerner method of property reduction, no other modelling methods were examined for porosity.  In short, this method was
seen to meet the goal of an acceptable method of modelling this type of manufacturing defect found in wind turbine blades.
Results in compression were similar for both cases, but were only considered moderate for the porosity case due to large
variation noted in the experimentation.

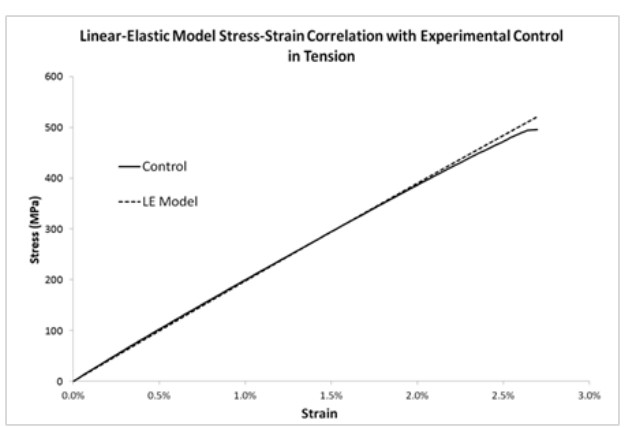 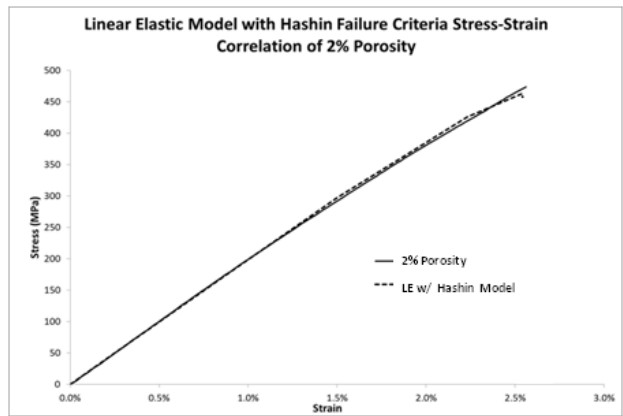

**Figure 8: Correlation of analytical and experimental results for the unflawed (left) and porosity (right) cases.**

## 4.2 Initial IP Wave Correlations

Assessment of the correlations from modelling the initial IP wave case resulted acceptance of the Hashin and combined
methods, but rejection of other methods (Table 5). A representative case comparing the as-tested IP wave with the combined
model results at similar displacements is shown below in Figure 9. The qualitative comparison was performed by comparing
the experimental images, taken from the data set shown with experimental stress-strain curve, at displacements of 0.5 mm and
2.0 mm with the model images generated at similar displacements. It is important to note by identifying these displacements
on the stress-strain curve, these snapshots along a similar progression. The reader is reminded that for the experimentation
full-field averages were used for strains (and for determining the material properties used), thus, a comparable approach was
used for modelled strain allowing for direct energy comparison. In Figure 9 (left), it may be seen that failure occurred first at
the edges where fibers were discontinuous at low loading which matches the degradation noted in both stress-strain curve. As
load increased, damage accumulation may be noted in the fiber misalignment section with shear failure occurring in the matrix
as the fibers straightened due to tensile elongation (Figure 9, right). As may be expected, the failure areas are cleaner and less
complex for the models due to uniformity and symmetry of the modelled specimen. For this case, the qualitative correlation
was quite consistent through the initial, low-load portion of each analysis where shear load increased significantly through the
wavy section for all the modelling techniques.

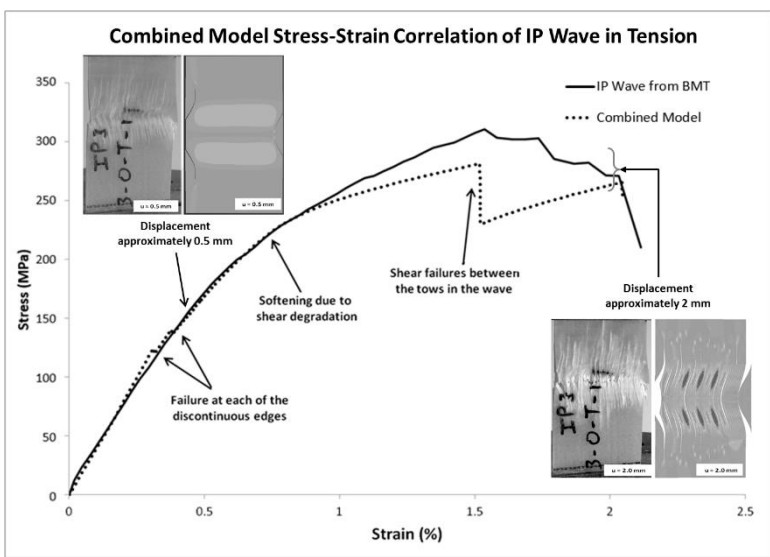

**Figure 9: Comparison of damage at displacements of approximately 0.5 mm and 2 mm between experimental (above) and analytical (below) showing onset-to-final damage left-to-right, respectively, with points and damage progression identified on resulting stress-strain curve.**

The resulting stress-strain curves from each model are shown in Figure 10. While the Hashin failure criteria successfully met the acceptance criteria, it did not exactly match the experimental material response particularly from 0.5-1.5% strain in tension (Figure 10). It is important to note that while the damage initiation parameters were not modified, the damage evolution parameters were modified to achieve the response shown. As noted in the methods and Table 4 above, the initial damage evolution parameters were approximated since they were not found experimentally. Given the acceptance criteria outlined above, the predicted strain at peak stress was 1.35% compared to 1.53% found experimentally or a variation of 12%. As such, these parameters were continually modified to 16e6 J/m$^2$, 16.9e6 J/m$^2$, 39.1e6 J/m$^2$, and 45.1e6 J/m$^2$, respectively for damage evolution parameters in Table 2. These values were used for the curve shown in Figure 10 as well as all additional modelling efforts. It is important to note that the damage evolution parameters are estimated from test data used to determine constitutive properties and ultimate strengths. Future work in pursuing this model would require individual constituent testing to determine the dissipation energies; however, given the intent of this paper, to compare various modelling methods, this was deemed to be outside the scope work. Correlation was only moderately acceptable in compression due to under-prediction of softening and over-prediction of final failure noted in Figure 10, right.

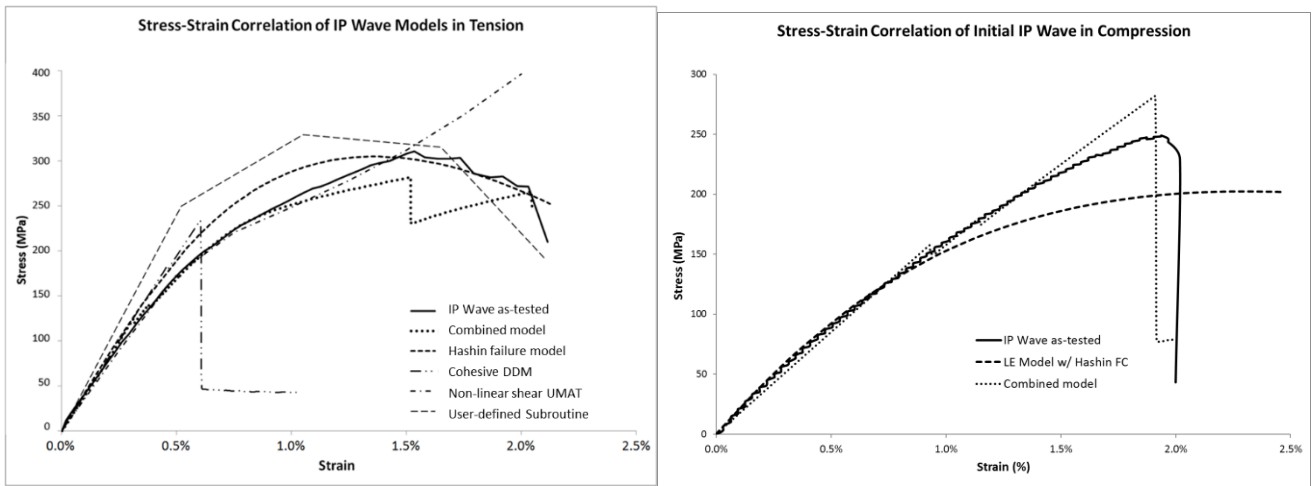

**Figure 10: Resulting initial IP Wave tension (left) and compression (right) stress-strain curves of each model compared to experimental results.**

In an unsuccessful attempt to offer the user more control to improve the modelling of the material response, the subroutine with user-defined failure criteria was used. As seen in Figure 10, the results in tension did not match the acceptance criteria even after modification of the damage initiation parameters within the accepted 10%. As seen in Figure 10 (left), the damage initiation began at approximately 1% strain where the peak stress was achieved. To achieve correlation, the damage initiation parameters would become unrealistic based on the experimentation. As such, this approach was deemed unacceptable, as noted in modulus correlation. and no further attempts at correlation were attempted. However, it does capture the overall shape, and if degradation of the initial modulus due to early progressive failure can be justified based on experimental validations, it may warrant further work.

While neither the non-linear shear subroutine nor using cohesive elements independently could accurately model the experimentally observed response, areas of promise were identified in tension. The non-linear shear response matched the experimental response up to failure more accurately than any other model (Figure 10) up to approximately 1.4% strain. At this point, the model showed the wavy fibers had essentially straightened resulting in the increased stiffness indicated. Given this was not seen experimentally, the approach was deemed unsuccessful. Similarly, when cohesive elements were placed between the fiber tows, matrix damage was modelled, though the peak stress and strain were both under-predicted. Since neither modelled the experimental damage progression, neither was used independently for additional cases (Table 5). Given these results, it was determined that adjusting the tabulated shear response or the traction-separation parameters, respectively, would have no useful impact on the failures of each of these models.

However, based on the individual responses of these two techniques, a model was created placing cohesive elements between the fibers of the non-linear shear subroutine model. When used to model the initial IP wave case the response correlated to the experimental data without any modification of the acceptable parameters. It follows that no modification would be necessary given the experimental nature of the shear response and the proven methodology of parametrically finding the

traction-separation parameters.  Specifically, the combined model curve and experimental IP wave curve had similar responses up to 0.5% strain as shown in Figure 10.  Above this point the model under-predicted the peak stress, which was attributed to the uniformity of the model which was based on the average fiber misalignment angle.  As such, the material failed through-the-thickness where all fibers were perfectly aligned, but the experimental specimens were not as consistent and some layers had a smaller fiber misalignment angle which increased the load carrying capability.  Regardless, the combined model was within the acceptable range and matched strain at failure where the cohesive failures caused the sudden drop in load-carrying capability.  Based on this result, and the moderate correlation in compression (Figure 10), additional correlations were attempted resulting in the best combination of accuracy, consistency, and predictive capability of all the modelling techniques tested (Table 5).

### 4.3 Initial OP Wave Correlations

Both the Hashin and combined models were run for the compression case (Figure 11).  Correlation was performed by comparing the full-field stress-strain data from the experimental to the model results.  As noted, no changes were made to the input model parameters from the tension cases above excepting the use of the compression data in Table 1.  Both models captured initial stiffness quite well up to approximately 1.5% strain where it is evident that the Hashin model was divergent resulting in only moderately acceptable correlation.  For the combined model, the first cohesive failures were noted at this strain though, load redistribution occurred and the model predicted the additional load-carrying before additional cohesive failures.  While the correlation was not perfect, it met the acceptance criteria.

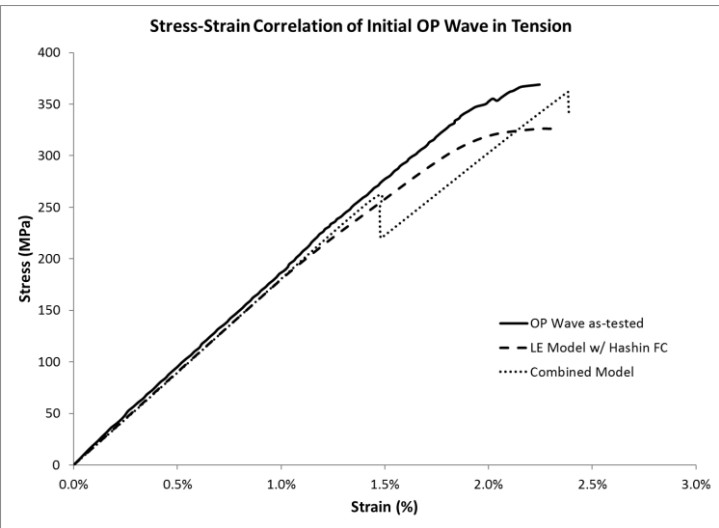

**Figure 11:  Resulting initial OP Wave stress-strain curves of each model compared to experimental results.**

## 4.4 Additional Wave Correlations

To match the experimental work, additional waves were modelled at 16° and 48° with no other changes made to any input parameters to assess the predictive capability of both the Hashin and combined approaches. For the Hashin approach, the initial case, over-predicted the load-carrying capacity after initially matching the stiffness, it very closely matched the appropriate stress at the ultimate failure strain. Neither of these bounding cases matched this result, but they both showed similar variations. The 16° case matched the initial stiffness and, similar to the initial case, over-predicted the load-carrying capacity before ultimately under-predicting the ultimate failure stress by just over 10%. The 48° case also matched the initial stiffness and over-predicted the load-carrying capacity. However, instead of being conservative this case also over-predicted the ultimate failure stress by almost 40%. Given only moderate results in compression above, the compression case was not run.

For the combined case, similar initial stiffness results were noted in both the 16° and the 48° IP wave cases. Instead of an over-prediction of the softening, a slight under-prediction was noted in the 16° case, while the 48° case appeared to match the overall softening quite well. As seen in Figure 12, the 16° case had an initial damage kink before softening began resulting in an under-prediction of peak stress of approximately 4.8%. The model then matched, within the same range, the continued load-carrying capacity up to truncation at failure of 2.0% strain. The 48° case delaminated in the discontinuous fiber sections, a significant number of cohesive elements failed resulting in a significant drop in load-carrying capacity, but not in final failure. As such the peak stress was noted at approximately 1% strain meaning correlation was unacceptable. Based on this result only the 16° combined model was correlated in compression. The results were very similar to the initial case shown in Figure 12 where stiffness was initially low for the model. In this case, the only one initial kink was noted and a stiffness change was associated with it. Unlike the initial case, the second kink occurred just before the peak stress which was over-predicted by approximately 5.1% with a predicted strain of 1.6% instead of the almost 1.8% observed experimentally.

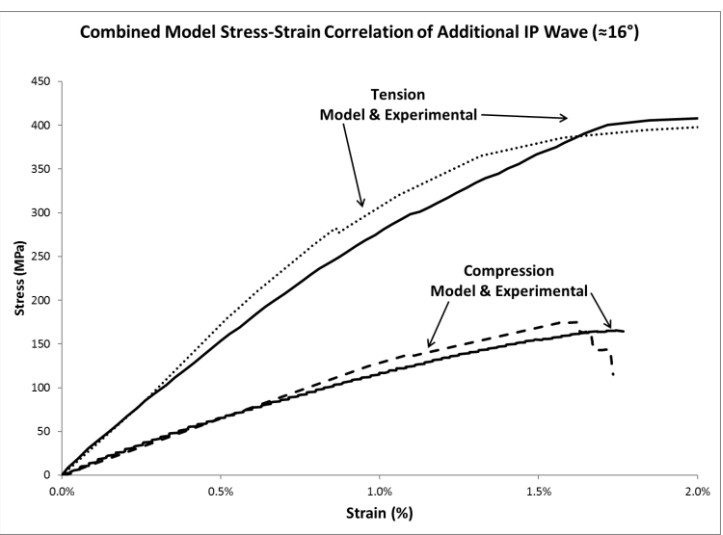

**Figure 12: Resulting additional IP Wave (16°) stress-strain curves in tension and compression compared to experimental results.**

## 5 Conclusions & Future Work

In summary, even though each model appeared to have different strengths, only the Hashin failure criteria and combined modelling techniques met the acceptable limits of the systematic approach employed. In both cases, this was true not only for the initial IP wave case, but also for additional wave and material cases. Going forward, the combined model more accurately predicted both the initial stress-strain response and damage, even though the computational time was five times longer. It is important to note that improved correlation may be possible with the Hashin based method given the approximations used for the damage evolution parameters. Since the combined approach was the found to be most the accurate, consistent, and predictive, such validation of the Hashin based method as implemented in ABAQUS was not performed due the need for constituent level experimentation which was beyond the scope of the work. Furthermore, while the Hashin model was adequate from a continuum mechanics sense, it does not physically represent the damage. That may be important in a damage tolerant design approach where damage inspections are necessary.

The application will dictate which approach, CDM or DDM, is most appropriate. If one only needs to know the global effects of local stiffness degradation due to damage, the CDM approach may be adequate. However, if one needs to know the actual damage, especially in a damage tolerant design, certification, and operating environment, integrating DDM may be useful. To assess and predict the effects of manufacturing defects common to composite wind turbine blades, a comparison of several different damage progression models was performed resulting in several conclusions. Findings indicate that when material properties generated from unflawed material testing were used, all models were able to predict initial laminate stiffness when flaw geometries are discretely modelled. Models were run. and a systematic approach was followed to assess the results compared to experimental results of flawed specimen. Specifically, the CDM using Hashin failure criteria was found to be accurate, consistent, and predictive in tension for all wave and material cases once the damage properties were found. However, even though it accurately predicted the stress strain response, it does not account for the actual, physical progressive damage observed during testing. To account for the variations noted and improve the accuracy, a user-defined failure criterion was run, but results were not within the acceptable limits. Next, non-linear shear UMAT and cohesive element approaches were independently developed and analysed. While each independently captured portions of the response, both resulted in unrealistic responses. However, when these two methods were combined, the result was the most accurate, consistent, and predictive correlation. It is important to note the significance of Table 5 which is a succinct evaluation of what to expect from the various models, and what needs to be improved for future work.

The results suggest these analytical approaches may be used to predict material response to possibly reduce material testing and traditional scalar safety factors, while also potentially supporting a probabilistic reliability and certification framework. For this to be achieved, future work emphasized on scalability is necessary to be sure local defects are considered as part of entire structure. This requires development of a multi-scale approach which requires an understanding of flaw response when surrounded by unflawed material. Appropriate modelling of this response will allow for a better understanding of flaws on larger structures.

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
