# Peer review of "Progressive Damage Modelling of Fiberglass/Epoxy Composites with Manufacturing Induced Waves Common to Wind Turbine Blades"

_Wind Energy Science, 2017_

## Referee Comment (RC1) · Anonymous Referee #1 · 15 May 2017

The manuscript adresses a relevant scientific question within the scope of WES but does not fulfill scientific quality. I suggest to rejected the manuscript in its current version.

Abstract: The abstract is fluffy and does not clearly state the work/scientific contribution. First the introduction shed some light on the work. The abstract does not provide a concise and complete summary and neither includes quantitative results.

Introduction: The introduction states the work tasks: Flaw characterization and effects of defects. The introduction does not give proper credit to related and relevant work in this area and does not at all quote any related work in the field of progressive damage in composites related to manufacturing induced waves beside from self-citations.

The differentiation between CDM and DDM is not clearly described. Furthermore, section 1.1 and 1.2 are textbook like copy-paste paragraphs from probably the PhD thesis and are not suited for a scientific article.

Section 3: Modelling Techniques: The section is very fluffy. The boundary conditions are not stated, instead referred to experimental work that is not described at all. The tables 1 and 2 are of bad quality.

Section 3.1 is a complete copy of the Abaqus manual.

Section 3.2 is ok, but does not really describe the user-defined subroutine well.

Section 3.4 Cohesive Zone Model. The statement "While previous convention was to utilize cohesive elements only in specific areas, pre-defining the crack path, computation availability has made it conceivable to place cohesive elements throughout the model. Thus, damage and crack progression may occur virtually anywhere in the model where the stress state indicates rather than where the user has placed these elements." is not correct. The damage and crack progression can still only occur where cohesive elements are placed. And since two cohesive elements cannot be connected and crack growth is limited to one direction, cracks can still only propagate inside the cohesive elements and not crack unification is possible.

Section 3.5 The authors state the inexact ability to determine the parameters for traction-separation laws. They might should read the publications from Bent F. Sørensen et al. Moreover, the parametric studies or any values are not shown/described.

Section 4.1 Figure 6: No legends are shown. Do the figures below in Figure 6 represent analytical or numerical results?

The entire results section is more quantitative, where initial models were tuned until a match was more or less reached. The manuscript does not provide any relevant scientific novel concepts, ideas, tools nor data.

---

## Referee Comment (RC2) · Anonymous Referee #2 · 6 Jun 2017

Overall, the writing is appreciated and the work contributes to a relevant area, however significant revisions should be made to the reporting of the work prior to being acceptable for publication.

The abstract should be rewritten with a more specific focus on the stated goal / hypothesis of the work and the conclusions clearly stated.

The introduction does not give proper credit to other research ongoing in the field and a more extensive literature review should be performed.

Boundary conditions of the models are not discussed yet are necessary for comparison to any experimental data, as well as for replication of the study.

There is no discussion of the test which is performed for validation. This should at least be touched on so it can be discussed within the context of the paper. I'm not sure what is being "compared" in figure 6. There is no scales or legends, and no actual data is shown for the experimental results. Perhaps this figure could be augmented to more clearly demonstrate what the author is trying to discuss.

No results are shown of testing other than the IP wave model. How did the other models compare with respect to OP models. No data is really talked about with respect to Porosity. If no models were run, why discuss it? If so, discuss the results.

---

## Author Comment (AC1) · 10 Jul 2017

**Author Comments for Review #1**

The authors are grateful for the comments, suggestions, and insight from the reviewer. The authors have made a significant effort to address all the comments below as we believe that not only is the question relevant, but the scientific quality is attainable given the work. We have revised the entire manuscript and believe it now surpasses the necessary scientific quality. Please find comments below [with text locations included where appropriate] and an updated version of the paper below.

**RC1** The manuscript adresses a relevant scientific question within the scope of WES but does not fulfill scientific quality. I suggest to rejected the manuscript in its current version.

    **AR** Significant revisions have been made and we respectfully request acceptance as we have addressed the reviewers' comments with this version.

**RC2** Abstract: The abstract is fluffy and does not clearly state the work/scientific contribution. First the introduction shed some light on the work. The abstract does not provide a concise and complete summary and neither includes quantitative results.

    **AR** Thank you for this comment. The abstract has been re-written to address these concerns. In particular, it has been re-written to focus on the scope of the content, and to orient the reader for expectations from the work.

**RC3** Introduction: The introduction states the work tasks: Flaw characterization and effects of defects. The introduction does not give proper credit to related and relevant work in this area and does not at all quote any related work in the field of progressive damage in composites related to manufacturing induced waves beside from self-citations.

    **AR** The introduction has been significantly modified to more clearly state the purpose and goal of this work. Additional references have been added throughout to cite the significant contributions of others. Given that the breadth of work in this area, the focus has been on foundational and influential works.

    Finally, the authors believe that the self-reference noted has been misconstrued as it refers to the companion paper that has been submitted with this paper that outlines the significant experimental work supporting the work presented here. Clarity of the importance of this companion paper and its distinction has been reinforced in the introduction, methods, and results.

**RC4** The differentiation between CDM and DDM is not clearly described. Furthermore, section 1.1 and 1.2 are textbook like copy-paste paragraphs from probably the PhD thesis and are not suited for a scientific article.

    **AR** These sections have been re-written to clearly define these techniques, emphasize the differences, and highlight the state of the art in these fields. Further clarifications have been made in the Conclusions and Future Work section.

**RC5** Section 3: Modelling Techniques: The section is very fluffy. The boundary conditions are not stated, instead referred to experimental work that is not described at all. The tables 1 and 2 are of bad quality.

**AR** The entire section has been reworked to identify the methods used and allow for recreation of the work described. Tables 1 and 2 have both been improved with additional descriptions and Figure 1 has been updated as well. [Note that this is now Section 2] Details and actual code have been added to provide the reader with all information necessary to reproduce the work if desired.

**RC6** Section 3.1 is a complete copy of the Abaqus manual.

    **AR** While the first sentence of this section continues to be a direct reference to this fact, the section has been truncated and the reader is referenced to Abaqus as needed. Only what is need to recreate the work is included. [Note that this is now Section 2.1] In particular, this is necessary to show the reader how different failure modes were modeled, and how the progressive damage with the Hashin model fits in with the Continuum Damage Modeling described in the paper.

**RC7** Section 3.2 is ok, but does not really describe the user-defined subroutine well.

    **AR** Additional clarity of the subroutine and user-defined failure criteria have been added. [Note that this is now Section 2.2]

**RC8** Section 3.4 Cohesive Zone Model. The statement "While previous convention was to utilize cohesive elements only in specific areas, pre-defining the crack path, computation availability has made it conceivable to place cohesive elements throughout the model. Thus, damage and crack progression may occur virtually anywhere in the model where the stress state indicates rather than where the user has placed these elements." is not correct. The damage and crack progression can still only occur where cohesive elements are placed. And since two cohesive elements cannot be connected and crack growth is limited to one direction, cracks can still only propagate inside the cohesive elements and not crack unification is possible.

    **AR** The authors understand the spirit of this comment and agree that our original statement is beyond the work described in here. Our group and others have made significant progress in development of intrinsic cohesive zone models where MPC's are selectively released allowing crack initiation and growth as described by the quoted statement above (https://dx.doi.org/10.4236/jamp.2014.212121). However, given that this is not descriptive of the work herein, we have chosen to reword these statements [now Section 2.4, p 12, lines 6-9]. Since the crack paths are generally known to be between fiber tows, the models herein have all of these potential paths (between every row of fibers) modeled at once instead of just a few given advances in computational availability. Further clarification is provided. "It is important to note that the damage does not necessarily occur at the cohesive zone area. It only provides the opportunity for growth where damage can, and has been experimentally determined to grow before final failure. Damage growth only occurs when and where the critical load is met. This is an important distinction from assuming a damage path as in the case of conventional Linear Elastic Fracture Mechanics."

**RC9** Section 3.5 The authors state the inexact ability to determine the parameters for traction-separation laws. They might should read the publications from Bent F. Sørensen et al. Moreover, the parametric studies or any values are not shown/described.

> **AR** A clearer depiction of the implementation of cohesive elements and the parametric study performed was added with apologies to Dr. Sørensen and others. In addition, description and a figure of the results of the parametric study have been added. [now section 2.5]

**RC10** Section 4.1 Figure 6: No legends are shown. Do the figures below in Figure 6 represent analytical or numerical results?

> **AR** Figure 6 has been reworked to more clearly show the correlations between the analytical and experimental work. Legends are not given for the strain fields of the analytical results because the authors believe they misrepresent the correlation. Instead a visual inspection shows similar damage progression. Addition of both stress-strain responses allows the reader to see not only the full-field average strain at these two points, but for entire loading. The authors appreciate this comment as it has led to this better explanation.

**RC11** The entire results section is more quantitative, where initial models were tuned until a match was more or less reached. The manuscript does not provide any relevant scientific novel concepts, ideas, tools nor data.

> **AR** As with the rest of the manuscript, significant changes have been made to ensure the novel scientific findings are apparent. Based on the reviewer's misunderstanding of how tuning took place, it is clear that the original manuscript did not adequate describe this process. It is worth noting that tuning only took place with the range of results seen the experimental results outlined in the companion paper. Input parameters were not arbitrarily modified to fit the data. They were soundly modified based on, and justified with, experimental and/or analytical data. It is understandable how the reviewer could not follow this path and the authors have ensured that this level of clarity exists throughout the manuscript.

[revised manuscript text omitted]

---

## Author Response (AR1)

**Author Comments for Review #1**

The authors are grateful for the comments, suggestions, and insight from the reviewer. The authors have made a significant effort to address all the comments below as we believe that not only is the question relevant, but the scientific quality is attainable given the work. We have revised the entire manuscript and believe it now surpasses the necessary scientific quality. Please find comments below [with text locations included where appropriate] and an updated version of the paper below.

**RC1** The manuscript adresses a relevant scientific question within the scope of WES but does not fulfill scientific quality. I suggest to rejected the manuscript in its current version.

    **AR** Significant revisions have been made and we respectfully request acceptance as we have addressed the reviewers' comments with this version.

**RC2** Abstract: The abstract is fluffy and does not clearly state the work/scientific contribution. First the introduction shed some light on the work. The abstract does not provide a concise and complete summary and neither includes quantitative results.

    **AR** Thank you for this comment. The abstract has been re-written to address these concerns. In particular, it has been re-written to focus on the scope of the content, and to orient the reader for expectations from the work.

**RC3** Introduction: The introduction states the work tasks: Flaw characterization and effects of defects. The introduction does not give proper credit to related and relevant work in this area and does not at all quote any related work in the field of progressive damage in composites related to manufacturing induced waves beside from self-citations.

    **AR** The introduction has been significantly modified to more clearly state the purpose and goal of this work. Additional references have been added throughout to cite the significant contributions of others. Given that the breadth of work in this area, the focus has been on foundational and influential works.

    Finally, the authors believe that the self-reference noted has been misconstrued as it refers to the companion paper that has been submitted with this paper that outlines the significant experimental work supporting the work presented here. Clarity of the importance of this companion paper and its distinction has been reinforced in the introduction, methods, and results.

**RC4** The differentiation between CDM and DDM is not clearly described. Furthermore, section 1.1 and 1.2 are textbook like copy-paste paragraphs from probably the PhD thesis and are not suited for a scientific article.

    **AR** These sections have been re-written to clearly define these techniques, emphasize the differences, and highlight the state of the art in these fields. Further clarifications have been made in the Conclusions and Future Work section.

**RC5** Section 3: Modelling Techniques: The section is very fluffy. The boundary conditions are not stated, instead referred to experimental work that is not described at all. The tables 1 and 2 are of bad quality.

    **AR** The entire section has been reworked to identify the methods used and allow for recreation of the work described. Tables 1 and 2 have both been improved with additional descriptions and Figure 1 has been updated as well. [Note that this is now Section 2] Details and actual code have been added to provide the reader with all information necessary to reproduce the work if desired.

**RC6** Section 3.1 is a complete copy of the Abaqus manual.

    **AR** While the first sentence of this section continues to be a direct reference to this fact, the section has been truncated and the reader is referenced to Abaqus as needed. Only what is need to recreate the work is included. [Note that this is now Section 2.1] In particular, this is necessary to show the reader how different failure modes were modeled, and how the progressive damage with the Hashin model fits in with the Continuum Damage Modeling described in the paper.

**RC7**   Section 3.2 is ok, but does not really describe the user-defined subroutine well.

    **AR**   Additional clarity of the subroutine and user-defined failure criteria have been added. [Note that this is now Section 2.2]

**RC8**   Section 3.4 Cohesive Zone Model. The statement "While previous convention was to utilize cohesive elements only in specific areas, pre-defining the crack path, computation availability has made it conceivable to place cohesive elements throughout the model. Thus, damage and crack progression may occur virtually anywhere in the model where the stress state indicates rather than where the user has placed these elements." is not correct. The damage and crack progression can still only occur where cohesive elements are placed. And since two cohesive elements cannot be connected and crack growth is limited to one direction, cracks can still only propagate inside the cohesive elements and not crack unification is possible.

    **AR**   The authors understand the spirit of this comment and agree that our original statement is beyond the work described in here.  Our group and others have made significant progress in development of intrinsic cohesive zone models where MPC's are selectively released allowing crack initiation and growth as described by the quoted statement above (https://dx.doi.org/10.4236/jamp.2014.212121).  However, given that this is not descriptive of the work herein, we have chosen to reword these statements [now Section 2.4, p 12, lines 6-9].  Since the crack paths are generally known to be between fiber tows, the models herein have all of these potential paths (between every row of fibers) modeled at once instead of just a few given advances in computational availability. Further clarification is provided.  "It is important to note that the damage does not necessarily occur at the cohesive zone area. It only provides the opportunity for growth where damage can, and has been experimentally determined to grow before final failure.  Damage growth only occurs when and where the critical load is met. This is an important distinction from assuming a damage path as in the case of conventional Linear Elastic Fracture Mechanics."

**Author Comments for Review #2**

The authors are grateful for the comments, suggestions, and insight from the reviewer. The authors have made a significant effort to address all the comments below as we believe that not only is the question relevant, but the scientific quality is attainable given the work. We have revised the entire manuscript and believe it now surpasses the necessary scientific quality. Please find comments below [with text locations included where appropriate] and an updated version of the paper below.

**RC1**      Overall, the writing is appreciated and the work contributes to a relevant area, however significant revisions should be made to the reporting of the work prior to being acceptable for publication.

        **AR**   Significant revisions have been made and we respectfully request acceptance of this major revision

**RC2**      The abstract should be rewritten with a more specific focus on the stated goal / hypothesis of the work and the conclusions clearly stated.

         Thank you for this comment. The abstract has been re-written with these comments in mind. In particular, it has been re-written to focus on the scope of the content, and to orient the reader for expectations from the work.

**RC3**      The introduction does not give proper credit to other research ongoing in the field and a more extensive literature review should be performed.

        **AR**   The introduction has been significantly modified to more clearly state the purpose and goal of this work. Additional references have been added throughout to cite the significant contributions of others. Given that the breadth of work in this area, the focus has been on foundational and influential works. The original manuscript included only foundational work, but other complementary work has been added for completeness.

**RC4**      Boundary conditions of the models are not discussed yet are necessary for comparison to any experimental data, as well as for replication of the study.

        **AR**   The entire section has been reworked to identify the methods used and allow for recreation of the work described. Tables 1 and 2 have both been improved with additional descriptions and Figure 1 has been updated as well. [Note that this is now Section 2] As for the experimental work, a companion paper has been submitted with this paper that outlines the significant experimental work supporting the work presented here. Clarity of the importance of this companion paper, and reference to it, has been reinforced in the introduction, methods, and results.

**RC5**      There is no discussion of the test which is performed for validation. This should at least be touched on so it can be discussed within the context of the paper. I'm not sure what is being "compared" in figure 6. There is no scales or legends, and no actual data is shown for the experimental results. Perhaps this figure could be augmented to more clearly demonstrate what the author is trying to discuss.

        **AR**   The entirety of what is now Section 3 [formerly Section 2] is a discussion of the validation test approach. It ties together the experimental data with the method of validation and what was considered acceptable modification of each model. This more effectively leads into the results and in particular sets up Table 6 [formerly Table 4] and Figure 9 [formerly Figure 6]. Figure 6 has been reworked to more clearly show the correlations between the analytical and experimental work. Legends are not given for the strain fields of the analytical results because the authors believe they misrepresent the correlation. Instead a visual inspection shows similar damage progression. Addition of both stress-strain responses allows the reader to see not only the full-field average strain at these two points, but for entire loading. The authors appreciate this comment as it has led to this better explanation.

***RC6*** No results are shown of testing other than the IP wave model. How did the other models compare with respect to OP models. No data is really talked about with respect to Porosity. If no models were run, why discuss it? If so, discuss the results.

    ***AR*** Specifics have been added for Porosity [Section 4.1], OP Waves [4.3], and additional waves [4.4] falling in line with expansion of results/discussion to deal with validation, tuning, etc.

**Progressive Damage Modelling of Fiberglass/Epoxy Composites with Manufacturing Induced Waves Common to Wind Turbine Blades**

Jared W. Nelson[1], Trey W. Riddle[2], and Douglas S. Cairns[3]

[1]SUNY New Paltz, Division of Engineering Programs, New Paltz, NY USA
[2]Sunstrand, LLC, Louisville, KY USA
[3]Montana State University, Dept. of Mechanical and Industrial Engineering, Bozeman, MT USA

**Abstract.** Composite wind turbne blades are typically reliable; owever, premature failures are often in regions of manufacturing defects. While the use of damage modelling has increased with improved computational capabilities, they are often performed for worst-case scenarios where damage or defects are replaced with notches or holes. To better understand and predict these effects, an effects of defects study has been undertaken. As a portion of this study, various progressive damage modelling approaches were investigated to determine if sufficient modelling capabilities existed to predict damage progression of composite laminates with typical manufacturing flawes included. ~~While the use of damage modelling has increased with improved computational capabilities, they are often performed for worst-case scenarios where damage or defects are replaced with notches or holes. To contribute to the establishment of a protocol understanding and quantifying the effects of these defects, a three round study was performed using continuum, discrete, and combined damage modelling. This approach relied ona.allyeingThese models were constructed to match the coupons from, and compare the results to, the characterization and material testing study. A standard defect case was chosen and initially used for each modelling approach to perform the qualitative and quantitative comparisons.It was found that while each model was able to show certain attributes, the most consistent, accurate, and predictive model was based on a combined continuum/discrete method.~~ Overall, the results indicate that this combined approach may provide insight into blade performance with known defects when used in conjunction with a probabilistic flaw framework.

**1 Introduction**

The US Department of Energy sponsored, Sandia National Laboratory led, Blade Reliability Collaborative (BRC) has been tasked with developing a comprehensive understanding of wind turbine blade reliability (Paquette, 2012).

A major component of this task is to characterize, understand, and predict the effects of manufacturing flaws commonly found in blades. Building upon coupon testing, outlined in the companion paper (Nelson et al., 2017), which determined material properties and characterized damage progression, three composite material defect types were investigated: porosity, in-plane (IP) waviness, and out-of-plane (OP) waviness. These defects were identified by an industry Delphi group as being common and deleterious to reliability (Paquette, 2012). Significant research into effects of common composite laminate defects has been performed for both porosity (Wisnom et al., 1996; Baley et al., 2004; Costa et al., 2005; Huang and Talreja, 2005; Pradeep et al., 2007; Zhu et al., 2009; Guo et al., 2009) and fiber waviness (Adams and Bell, 1995; Adams and Hyer, 1996; Cairns et al., 1999; Niu and Talreja, 1999; Avery et al., 2004; Wang et al., 2012; Lemanski et al., 2013; Mandell and Samborsky, 2013). The goal of this portion of the overall project was to establish analytical approaches to model progressive damage in flawed composite laminates consistently and accurately predict laminate response. Multiple cases for each flaw type were tested allowing for progressive damage quantification, material property definition, and development of many correlation points in this work. As outlined in the following sections, there have been two primary modelling approaches used to assess damage progression in composite laminates: Continuum Damage Modelling (CDM), and Discrete Damage Modelling (DDM). While these methods are well established, there has been little work directly assessing predictive capabilities when applied to wind turbine blade laminates with defects.

A major component of this task is to characterize and understand manufacturing flaws commonly found in blades. In this paper, the authors describe and develop two tasks; Flaw Characterization and Effects of Defects. Characterizing flaws is necessary to determine and quantify what manufacturing defects are present. The Effects of Defects is focused on understanding the mechanical performance of materials containing typical flaws and comparing various progressive damage models techniques. Different analytical approaches to model progressive damage in flawed composite laminates for consistency, accuracy, and predictive capability were developed and evaluated.. Building upon coupon testing which determined material properties and characterized damage progression, three composite material defect types were investigated: in-plane (IP) waviness, out-of-plane (OP) waviness, and porosity (Nelson et al., 2017). Multiple cases for each flaw type were tested allowing for progressive damage quantification, material property definition, and development of many correlation points in this work.

**1.1 Continuum Damage Modelling Background**

Two distinct modelling methods have been investigated and compared: Continuum Damage Modelling (CDM), and Discrete Damage Modelling (DDM) (Nelson et al., 2012; Woo et al, 2013). Continuum Damage Modelling (CDM) is a "pseudo-representation" that does not explicitly model the exact damage but instead, updates the constitutive properties as damage

occurs (Kachanov, 1986).  This allows for the relating of equations to heterogeneous micro-processes that occur during strain of materials locally, and during strain of structures globally, insofar as they are to be described by global continuum variables given their non-homogeneity (Talreja, 1985; Chaboche, 1995). ~~Simply put, actual description of damage is difficult, especially when the damage is on a grain, cell, or micro scale; however, a change in global material response is rather easily noted from the onset of damage. As such, it is often useful to homogenize the material properties of the RVE, a process which is not always feasible when studying composites. In some instances, the two different materials cannot be represented accurately in such a way, especially when damage occurs independently in one of the constituents. Thus, care must be given to the failure modes and types when accounting for the changes in constitutive properties through the damage phases.~~

Thus, for typical CDM as the model iterates at each strain level, the constitutive matrix is updated to reflect equilibrium damage. Then as damage occurs, the elastic properties are irreversibly affected in ways that are similar to those in a general framework of an irreversible thermodynamic process (Kachanov, 1986). This may takes place by reducing the elastic properties ($E_1$, $E_2$, $v_{12}$, $v_{32}$, and $G_{12}$) in the stiffness matrix ($C$) of the stress-strain relationship. Damage is not directly measurable from this approach, but may be estimated for the continuum by altering observable properties: strength, stiffness, toughness, stability, and residual life.

$$
\begin{Bmatrix} \sigma_{xx} \\ \sigma_{yy} \\ \sigma_{zz} \\ \sigma_{yz} \\ \sigma_{zx} \\ \sigma_{xy} \end{Bmatrix} =
\begin{Bmatrix}
C_{11} & C_{12} & C_{13} & 0 & 0 & 0 \\
 & C_{11} & C_{13} & 0 & 0 & 0 \\
 & & C_{33} & 0 & 0 & 0 \\
 & & & C_{44} & 0 & 0 \\
 & sym & & & C_{44} & 0 \\
 & & & & & (C_{12} - C_{12})/2
\end{Bmatrix}
\begin{Bmatrix} \varepsilon_{xx} \\ \varepsilon_{yy} \\ \varepsilon_{zz} \\ \varepsilon_{yz} \\ \varepsilon_{zx} \\ \varepsilon_{xy} \end{Bmatrix}
$$

~~Continuum approaches for composite materials have been well established (Blacketter et al., 1993; Chapman and Whitcomb, 2000; Gorbatikh et al., 2007). In some cases, finite element analysis has been used to account independently for fiber and matrix damage. Chang and Chang (1987) developed a composite laminate in tension with a circular hole where material properties were degraded to represent damage. Failure criteria were defined based on the failure mechanisms resulting from damage: matrix cracking, fiber-matrix shearing, and fiber breakage. A property reduction model was implemented and the results were in agreement for seven (7) independent laminates. Later,~~ Damage may be viewed as the creation of discontinuities, and Kachanov (1986) suggested a single scalar variable as a measure of the effective surface density of these discontinuities. This approach assumes that load redistribution to undamaged areas or ligaments occurs, and effective stresses increase, until

all ligaments are severed at failure.  This tensorial representation can take any direction in the continuum, but it must be expanded to at least a second order tensor for utility of an orthotropic material, or to a fourth order tensor to generalize damage through elimination of any material symmetry (Chaboche, 1995; Cauvin and Testa, 1999; Carol et al., 2001; Luccioni and Oller, 2003; Maimi et al, 2007).

5   There are two crucial considerations when modelling damage: the failure theory and ways to account for the damage. Typical failure criteria such as the maximum stress, the maximum strain, Hashin (1981), Tsai–Hill (1968), and Tsai–Wu (1971) are widely used because they are simple and easy to utilize (Christensen, 1997).  In reviews by Daniel (2007) and Icardi (2007), wide variations in prediction by various theories were attributed to different methods of modelling the progressive failure process, the non-linear behavior of matrix-dominated laminates, the inclusion or exclusion of curing residual stresses in the

10   analysis, and the utilized definition of ultimate failure.  Camanho and Matthews (1999) achieved reasonable experimental/analytical correlation using Hashin's failure theory to predict damage progression and strength in bearing, net-tension, and shear-out modes.

  To account for damage, progressive damage models of composite structures range from the simple material property degradation methods (MPDM) to more complex MPDM that combine CDM and fracture mechanics (Tay et al., 2005).

15   Implementing a ply discount method whereby the entire set of stiffness properties of a ply is removed from consideration if the ply is deemed to have failed has been well established (Pal and Ray; 2002; Prusty, 2005; Maimi et al., 2007).  Studies have also been performed attempting to generalize this procedure by replacing the constant degradation factor with a gradual stiffness reduction scheme (Reddy 
[revised manuscript text omitted]

**4.3 Initial OP Wave Correlations**

HASHIN…. Both the Hashin and combined models The model was alsowere run for the compression case (as shown in Figure 68, right) and while the compression case seemed reasonable, there are no data for comparison, and so no correlation was made; however, the model is ready when future data become available.

COMBINED…As above, quantitative cCorrelation was performed by comparing the full-field stress-strain data from the BMTexperimental to the model results. As noted, no changes were made to the input model parameters other than the change in model geometry. TheBoth models captured initial stiffness quite well up to approximately 1.5% strain where it is clearly evident that the Hashin model was divergent resulting in only moderately acceptable correlation. For the combined model at which point the first cohesive failures were noted at this strain though, which aligned with the transition from Figure 108a b. lLoad redistribution occurred and the model predicted the additional load-carrying seen through Figure 108cbefore additional cohesive failures and manual truncation occurred. While the correlation was not perfect, it met the acceptance criteria. matched fairly well. Tuning may be applicable as future work with additional testing and especially with improved understanding of response when this flaw is embedded into a substructure or structure.

[Figure]

**Figure 11: Resulting initial OP Wave stress-strain curves of each model compared to experimental results.**

In summary, even though each model appeared to have different strengths, only the Hashin failure criteria and combined modelling techniques met the acceptable limits of the systematic approach employed. In both cases, this was true not only for the initial IP wave case, but also for additional wave and material cases (Table 6Table 4). After initial tuning of the damage parameters, the Hashin failure criteria model showed acceptable correlation in tension and moderate correlation in compression, while the combined model was acceptable for both. In tension, the combined model more accurately predicted both the initial stress-strain response and damage, even though the computational time was five times longer. when considering all cases, the combined approach was the found to be most the accurate, consistent, and predictive.

**4.4 Additional Wave Correlations**

HASHIN…To match the experimental work, additional waves were modelled at 16° and 48° with no other changes made to any input parameters to assess the predictive capability of both the Hashin and combined approaches. For the Hashin approach, While the initial case above, over-predicted the load-carrying capacity after initially matching the stiffness, it very closely matched the appropriate stress at the ultimate failure strain. Neither of these bounding cases matched this result, but they both showed similar variations (Figure 70). The 16° case matched the initial stiffness and, similar to the initial 29° case, over-predicted the load-carrying capacity before ultimately under-predicting the ultimate failure stress by just over 10%. The 48° case also matched the initial stiffness and over-predicted the load-carrying capacity. However, instead of being conservative this case also over-predicted the ultimate failure stress by almost 40%. Given only moderate results in compression above, the compression case was not run.…

COMBINED….For the combined case Similar initial stiffness results were noted in both the 16° and the 48° IP wave cases. Instead of an over-prediction of the softening, a slight under-prediction was noted in the 16° case, while the 48° case appeared to match the overall softening quite well. As seen in Figure 12-12Figure 113, the 16° case had an initial damage kink before softening began resulting in an under-prediction of peak stress of approximately 4.8%. The model then matched, within the same range, the continued load-carrying capacity up to truncation at failure of 2.05% strain. The 48° case told a different story and saw the best correlation of any model case in this entire study through softening. However, shortly after the kinks associated with the delaminateding in theof discontinuous fiber sections, a significant number of cohesive elements failed resulting in a significant drop in load-carrying capacity, but not in final failure. As such the peak stress was noted at approximately 1% strain and was found to be almost 20% below the peak stress noted in the model at 1.7% strainmeaning correlation was unacceptable. After this peak, however, a slight increase in load-carrying capacity was noted up to truncation at failure of 2.4% strain. As noted abovBased on this result e-only the 16° combined model yieldedwas correlated useful results in compression. The results were very similar to the initial case shown in Figure 12-12Figure 104 where stiffness was initially slightly-low for the model and was again attributed to variation in the BMT coupons tested. 
[revised manuscript text omitted]

---

## Author Response (AR2)

Additional changes have been made and significant clarification has been made to ensure reproducibility. We have chosen to address these in the body of the paper, as we feel this will clarify the work performed and analysis of the results. Addressing several specific issues (references are giving in [page:line] format and refer to document below):

- Better clarification to guide reader through Table 4 and the related portion of Section 3 which directly addresses the methods of parameter tuning. Further, this has been identified throughout the paper to ensure the reader clearly understands and is able to reproduce the results. [6:10-23 8:3-6; 10:23-36; 12:6-7]

- In particular, additional clarity has been added throughout to overcome the impression that the model inputs were changed to achieve acceptable correlation [5:1-4; 8:3-6; 10:1-2; 10:23-26; Table 3; 11:7; 12:6-7; 19:7-16; 20:6-8; 20:18-20; 20:23-24; 21: 13-14; 22:1-2]

- Additional text in the results section has been added to clarify the result finding and explanation to interpret why some models worked, and others did not, has been extended. [19:7-16; 20:6-8; 20:18-20; 20:23-24]

- It is important to note that not all values have been put into Tables in this manuscript. For example, the peak stress used in the cohesive law has been in all versions so far (p. 11, line 21) and are followed by the mixed-mode criterion inputs. However, a Table has been added with the shear points used in Figure 3.

- Based on initial comments of the reviewer, the Hashin based model approach was dramatically reduced; too much so in hindsight. Additional explanation has been added, including the method of determination for the damage evolution parameters. As such, discussion of these values follows through into the discussion of the results and allows the reader to reproduce them if desired. [6:10-23; Table 2; 19:7-16; 23: 5-11]

While the authors agree with the spirit and intent of the Reviewer #1's comments and suggestions, it does appear that there is a pre-conceived impression that parameters were changed to match experimental results. This is not the case. All parameters that were available to be modified (e.g. those clearly identified in Sections 2.1-2.5 and summarized in Table 4), were chosen because either the experimental work (e.g. accompanying wes-2017-13) showed some variation or approximations were made because data were not available (e.g. Hashin damage evolution parameters). Both are now more clearly identified throughout and their significance addressed.

Finally, we are appreciative of the feedback and recognition of from both reviewers on the improvement of this work. We also appreciate the effort of the editors to ensure the paper is presented correctly.

A marked-up version of the paper with comments highlighting the changes follows:

[revised manuscript text omitted]

**Commented [n5]:** Additional text to clarify the modifications available for the shear model.

[Figure]

**Commented [n6]:** Updated figure to include tabulated shear points used in the non-linear shear model for user re-creation

**Figure 3: Key points from empirical shear stress-strain relationship used in the non-linear shear UMAT.**

**2.4 Cohesive Zone Model**

To model damage progression discretely, cohesive elements are typically utilized based on a cohesive law relating traction to separation across the interface (Karayev et al., 2012; Lemanski et al., 2013). Zero thickness elements with specific bi-linear traction-separation criteria (Figure 5, right) were placed between the fiber tows of the material properties in Table 1 above. While following convention to utilize cohesive elements only in specific areas, computational availability has made it conceivable to place cohesive elements between all fiber tows throughout the model. Thus, damage and crack progression could occur between any fibers based on the stress state. It is important to note that the damage does not necessarily occur at the cohesive zone area. It only provides the opportunity for growth where damage can, and has been experimentally determined to grow before final failure. Damage growth only occurs when and where the critical load is met. This is an important distinction from assuming a damage path as in the case of conventional Linear Elastic Fracture Mechanics.

A bi-linear traction-separation criterion was implemented (Figure 4) where the initial stiffness, $K$, of the cohesive element is linear up to the damage initiation point at critical separation, $\Delta_c$. From this point to the failure separation, $\Delta_{fail}$, the slope estimates the damage evolution of each the cohesive element up to failure. The traction-separation criterion is met for a specific cohesive element and a separation occurs resulting in crack propagation and element deletion. A standard material specification was used and parametric studies were performed to determine the stiffness and maximum traction properties of the cohesive elements (Figure 5). Given these parametric studies, it was deemed that these values may be modified if necessary within the ranges determined from the study. Initial model analyses were performed to determine the cohesive element stiffness, $K_{eff}$. Analyses were performed at various stiffness values to determine the convergence value of 5E6 N/mm by performing several model runs to determine convergence point (Figure 5a). Similarly, the effects of $T_{1max}$ were determined by analyzing several different values and it was determined that failure behavior was not dependent on $T_{1max}$ (Figure 5b). However, when a similar test was analyzed for $T_{2max}$ it was quickly apparent that the failure was sensitive to Mode II shear damage (Figure 5c). The peak tractions ($T_{1max} = T_{2max} = 100$ MPa) were then used in an initial run as shown in Figure 5d to confirm these values and ensure that the cohesive elements were not influencing initial stiffness correlations. A B-K mixed mode criterion was utilized where $G_{Ic}$ and $G_{IIc}$ were 806 J/m$^2$ and 1524 J/m$^2$, respectively, as found experimentally (Benzeggagh and Kenane, 1996).

**Commented [n7]:** Clarification of available modification for the cohesive model approach

**Commented [n8]:** Peak tractions used throughout

[revised manuscript text omitted]